# Persistence on oral pre-exposure prophylaxis (PrEP) among female sex workers in eThekwini, South Africa, 2016–2020

Amrita Rao[1]*, Hlengiwe Mhlophe[2], Carly Comins[1], Katherine Young[3], Mfezi Mcingana[3], Catherine Lesko[1], Ntambue Mulumba[2], Stefan Baral[1], Harry Hausler[3], Sheree Schwartz[1]

1 Department of Epidemiology, Johns Hopkins Bloomberg School of Public Health, Baltimore, Maryland, United States of America, 2 TB HIV Care, Durban, South Africa, 3 TB HIV Care, Cape Town, South Africa

* arao24@jhu.edu

## Abstract

### Background

Despite the established efficacy of PrEP to prevent HIV and the advantages of a user-controlled method, PrEP uptake and persistence by women in both trials and demonstration projects has been suboptimal. We utilized real-world data from an HIV service provider to describe persistence on oral PrEP among female sex workers (FSW) in eThekwini, South Africa.

### Methods

We examined time from PrEP initiation to discontinuation among all FSW initiating PrEP at TB HIV Care in eThekwini between 2016–2020. We used a discrete time-to-event data setup and stacked cumulative incidence function plots, displaying the competing risks of 1) not returning for PrEP, 2) client discontinuation, and 3) provider discontinuation. We calculated hazard ratios using complementary log-log regression and sub-hazard ratios using competing risks regression.

### Results

The number of initiations increased each year from 155 (9.3%, n = 155/1659) in 2016 to 1224 (27.5%, n = 1224/4446) in 2020. Persistence 1-month after initiation was 53% (95% CI: 51%-55%). Younger women were more likely to discontinue PrEP by not returning compared with those 25 years and older. Risk of discontinuation through non-return declined for those initiating in later years. Despite the COVID-19 pandemic, a greater number of initiations and sustained persistence were observed in 2020.

### Conclusions

Low levels of PrEP persistence were observed, consistent with data among underserved women elsewhere. Encouragingly, the proportion of women persisting increased over time, even as the number of women newly initiating PrEP and staff workload increased. Further

**Data Availability Statement:** This analysis leveraged program data that the research team does not have ownership over. Data are protected under the Protection Of Personal Information Act

(POPI) in South Africa. De-identified data may be made available by request from TB HIV Care in accordance with national laws and organizational regulations. Requests for de-identified data can be sent to Anje Pretorius at Anje@tbhivcare.org.

**Funding:** AR;F31MH124458-01A1; National Institute of Mental Health; https://nam02.safelinks. protection.outlook.com/?url=https%3A%2F% 2Fwww.nimh.nih.gov%2F&data=04%7C01% 7Carao24%40jhu.edu%7C0cbfd826678f 462fe52c08d9fd5835b4%7C9fa4f438b1e 6473b803f86f8aedf0dec%7C0%7C0% 7C637819378360948543%7CUnknown% 

research is needed to understand which implementation strategies the program may have enacted to facilitate these improvements and what further changes may be necessary.

## Introduction

Nearly one million people are newly infected with HIV annually in South Africa, constituting about 15% of global incident infections [1]. Female sex workers (FSW) bear a disproportionate burden of disease in the country. Estimates suggest that 60% of cisgender FSW are living with HIV compared with 19% of other adults of reproductive age [1–4]. FSW experience high levels of violence from clients, non-paying partners, sex work managers, and others [5,6]. This coupled with arrest, economic necessity to accept unsafe work conditions, and other power imbalances often compromise their ability to negotiate consistent condom use [7–9]. Pre-exposure prophylaxis for HIV prevention (PrEP) is an efficacious and user-controlled option to prevent new HIV infections [10,11]. Daily oral PrEP can be used autonomously and discreetly, without the need of partner or client involvement, and presents an opportunity for women to protect themselves from HIV [12,13].

Despite the HIV protection conferred by PrEP, with some estimates suggesting that daily use reduces acquisition by 79–85% among women [10,11], PrEP persistence among FSW in trial settings and demonstration projects has been low [14,15]. Even if freely available, many factors influence a woman's decision to initiate and continue taking PrEP, including knowledge, side-effects, and accessibility of PrEP, along with actual and perceived risk of HIV acquisition, and PrEP stigma and social influence [16–19].

South Africa became the first country in Sub-Saharan Africa to implement PrEP as part of their national strategy for HIV prevention and began providing PrEP to FSW in 2016 [20]. PrEP has been primarily delivered through specific programs, which have existing HIV services in place and established rapport with community groups. The objective of this analysis was to describe PrEP persistence among cisgender FSW accessing PrEP in real-world conditions in eThekwini, KwaZulu-Natal from 2016 to 2020.

## Methods

To describe PrEP persistence among FSW, we utilized data collected by TB HIV Care as part of routine service delivery. TB HIV Care is a South African non-profit organization that serves as the country's largest PrEP provider for young women and FSW and began providing PrEP to FSW in 2016. They now operate at multiple sites across five provinces and offer services through mobile van health clinics and drop-in center wellness clinics.

The study population for this analysis includes cisgender FSW assigned female sex at birth who initiated PrEP through TB HIV Care in eThekwini between September 1st, 2016, and December 31st, 2020. Women would have been eligible to begin PrEP through TB HIV Care if they were already accessing prevention services (e.g. testing for pregnancy, STIs, or HIV; family planning) through the program serving FSW and were HIV-negative.

Data for this study come from a site-level register that is maintained to track PrEP uptake and persistence over time and manage patient follow-up and scheduling. This register has individual-level data on the date of first PrEP initiation and the outcomes of subsequent monthly visits, including whether or not individuals stopped and restarted PrEP. We followed women from PrEP initiation to discontinuation or administrative censoring at 12 months or December 2020, whichever came first. Discontinuation was defined as a composite outcome

based on 1) not attending two consecutive visits (not returning for PrEP), 2) attending a monthly appointment and the client choosing to discontinue PrEP (client discontinuation), or 3) attending a monthly appointment and the clinician deciding to discontinue PrEP (provider discontinuation). Two consecutive missed visits was used to define discontinuation in alignment with TB HIV Care's PEPFAR reporting definition for loss to follow-up. If a client misses one visit, but returns the next month, the program notes this client as not attending a visit but continuing on PrEP. If a client misses two consecutive visits and returns subsequently, the program defines her as a "restart."

We examined time-to-PrEP discontinuation using a discrete time-to-event data setup, with person-periods defined for each month in which a woman was observed. We refer to the complement of PrEP discontinuation (i.e. 'survival' from discontinuation) as PrEP persistence. We plotted stacked cumulative incidence functions to show the differential risk of types of PrEP discontinuation (client discontinuation, provider discontinuation, not returning for PrEP) over time. We stratified analyses by age and era-of-initiation, defined as the year in which a woman initiated PrEP (2016–2020). We calculated overall, "cause-specific" hazard ratios (CHR) for discontinuation by age and year of initiation using a complementary log-log regression model. We calculated subdistribution-hazard ratios (SHR) for each specific type of PrEP discontinuation using Fine and Gray survival models [21]. Results were determined to be statistically significant if the 95% confidence interval around the hazard ratio did not overlap with the line of no difference, which is 1 in the case of ratio measures. All analyses were conducted using Stata 14.2 (StataCorp, College Park, Texas, USA).

## Results

There were a total of 2776 FSW newly initiating PrEP at TB HIV Care in eThekwini between 2016 and 2020 and included in these analyses. Approximately 60% were women ages 25 years and older (1662/2762). The absolute number of PrEP initiations increased over time, and the percentage of PrEP initiations among those who were seen by the program and HIV-negative increased substantially in 2019 and 2020: 155/1659 (9.3%) in 2016, 211/3235 (6.5%) in 2017, 428/6042 (7.1%) in 2018, 756/5789 (13.1%) in 2019, and 1224/4446 (27.5%) in 2020 (Fig 1). Based on the Kaplan-Meier survival function, PrEP persistence was 53% (95% CI: 51%-55%) 1-month after initiation, 33% (95% CI: 31%-35%) 4-months after initiation, and 18% (95% CI: 16%-19%) 7-months after initiation. By 12-months, 9% (95% CI: 7%-10%) persisted on PrEP. Details of the Kaplan-Meier survivor function are provided in the S1 Appendix. Among those who discontinued (n = 2210), 13% restarted during the observation period (287/2210), with a median time from discontinuation to restart of 3 months (IQR: 3–5).

At 1-month after initiation, 56% (95% CI: 54%-59%) of women ≥25 years old persisted on PrEP compared with 48% (95% CI: 45%-51%) of women <25 years old. The higher PrEP persistence seen in women ≥25 years was predominantly due to better attendance at follow-up visits compared with women <25 years (SHR for program-defined loss to follow-up: 0.82, 95% CI: [0.76, 0.88]). There were no significant differences by age in client-initiated or provider-initiated discontinuation (Table 1).

PrEP persistence improved over the course of the program (Fig 2). In 2019, 62% of women persisted on PrEP 1-month after initiation (95% CI: 58%-65%). By comparison, in 2016, 52% of women persisted on PrEP 1-month after initiation (95% CI: 43%-59%). Women initiating PrEP in 2018 (SHR: 0.84, 95% CI: [0.72, 0.99]), 2019 (SHR: 0.64, 95% CI: [0.55, 0.75]) and 2020 (SHR: 0.82, 95% CI: [0.71, 0.95]) were statistically significantly less likely to stop coming back for their PrEP compared to those initiating in 2016, when accounting for the competing risks of client-initiated and provider-initiated discontinuation. Interestingly, there was a

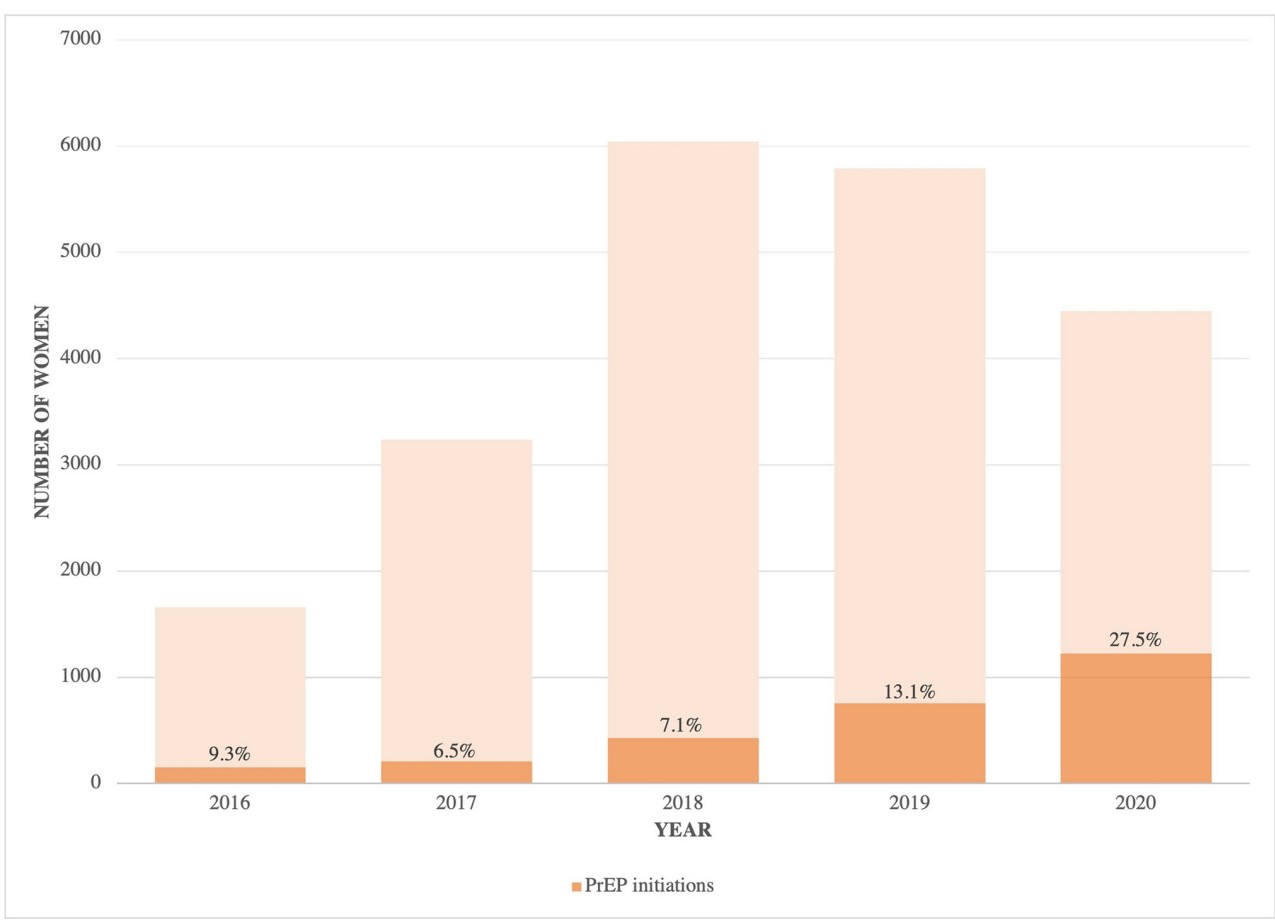

**Fig 1. Number of PrEP eligible (HIV-negative and accessing services through TB HIV Care) and number and percentage of PrEP initiations each year among female sex workers in Durban, South Africa, 2016–2020.**

statistically significantly greater risk of client-initiated discontinuation in 2019 compared with 2016, with 150 client-initiated discontinuations out of the 756 who initiated in 2019, or 19.8%, (SHR: 3.41, 95% CI: [1.75, 6.63]) (Table 1).

## Discussion

Among FSW accessing HIV prevention services from a real-world service provider in eThekwini, South Africa, the number of PrEP initiations increased each year from 2016 to 2020. PrEP persistence declined to 53% one month after initiation and to just 9% after one year. Younger women were more likely to discontinue PrEP due to missed visits compared with those 25 years and older. Risk of discontinuation due to missed visits declined for those initiating in 2018, 2019, and 2020, suggesting improvements over the life of the program in helping women to stay engaged. Despite the disruptions to the healthcare infrastructure created by the COVID-19 pandemic, the program reported more initiations, sustained improvements in persistence, and a lower risk of client-initiated stops among those initiating in 2020.

Poor PrEP persistence may challenge its real-world utility. The low levels of persistence seen in this study and the sharp decline at one-month are consistent with what has been reported in trials, including the FEM-PrEP study [22] and VOICE [23]. The *TAPS*

**Table 1. Hazard of PrEP discontinuation and sub-hazard ratios of two consecutive missed visits, client-initiated discontinuation, and provider-initiated discontinuation among 2776 female sex workers initiating PrEP through TB HIV Care in Durban, South Africa 2016–2020.**

| | UNIVARIABLE | | | | |
|---|---|---|---|---|---|
| | Number of initiations | Discontinuation (composite outcome) | Two consecutive missed visits (n = 1986) | Client-initiated discontinuation (n = 217) | Provider-initiated discontinuation (n = 7) |
| | | Hazard ratio (95% CI) | Sub-hazard ratio (95% CI) | | |
| **Age**[1] | | | | | |
| <25 years old | 1102 | ref | ref | ref | ref |
| 25+ years old | 1662 | **0.80 (0.74, 0.86)** | **0.84 (0.79, 0.91)** | 0.97 (0.74, 1.26) | 0.91 (0.20, 4.10) |
| **Year**[2] | | | | | |
| 2016 | 155 | ref | ref | ref | ref |
| 2017 | 211 | 0.99 (0.82, 1.19) | 0.96 (0.80, 1.16) | 0.89 (0.37, 2.14) | 0.73 (0.05, 11.72) |
| 2018 | 428 | 0.84 (0.71, 1.00) | 0.86 (0.73, 1.01) | 1.02 (0.48, 2.16) | 1.41 (0.16, 12.56) |
| 2019 | 756 | 0.88 (0.75, 1.03) | **0.67 (0.57, 0.78)** | **3.41 (1.75, 6.63)** | 0.20 (0.01, 3.09) |
| 2020 | 1224 | **0.82 (0.70, 0.96)** | **0.83 (0.71, 0.96)** | **0.35 (0.16, 0.77)** | *Too few events* |
| | INCLUDING BOTH AGE AND YEAR IN THE SAME MODEL | | | | |
| | | Hazard ratio (95% CI) | Sub-hazard ratio (95% CI) | | |
| **Age**[1] | | | | | |
| <25 years old | 1102 | ref | ref | ref | ref |
| 25+ years old | 1662 | **0.80 (0.74, 0.86)** | **0.82 (0.76, 0.88)** | 1.10 (0.85, 1.43) | 0.85 (0.18, 3.98) |
| **Year**[2] | | | | | |
| 2016 | 155 | Ref | ref | ref | ref |
| 2017 | 211 | 0.96 (0.80, 1.16) | 0.94 (0.78, 1.12) | 0.90 (0.38, 2.16) | 0.72 (0.04, 11.67) |
| 2018 | 428 | 0.83 (0.70, 0.98) | **0.84 (0.72, 0.99)** | 1.02 (0.48, 2.17) | 1.40 (0.15, 13.10) |
| 2019 | 756 | 0.85 (0.72, 0.99) | **0.64 (0.55, 0.75)** | **3.46 (1.78, 6.71)** | 0.19 (0.01, 3.47) |
| 2020 | 1224 | **0.82 (0.70, 0.96)** | **0.82 (0.71, 0.95)** | **0.36 (0.16, 0.78)** | *Too few events* |

[1] Age data missing for n = 12 (0.4%);

[2] Year of initiation data missing for n = 2 (0.07%).

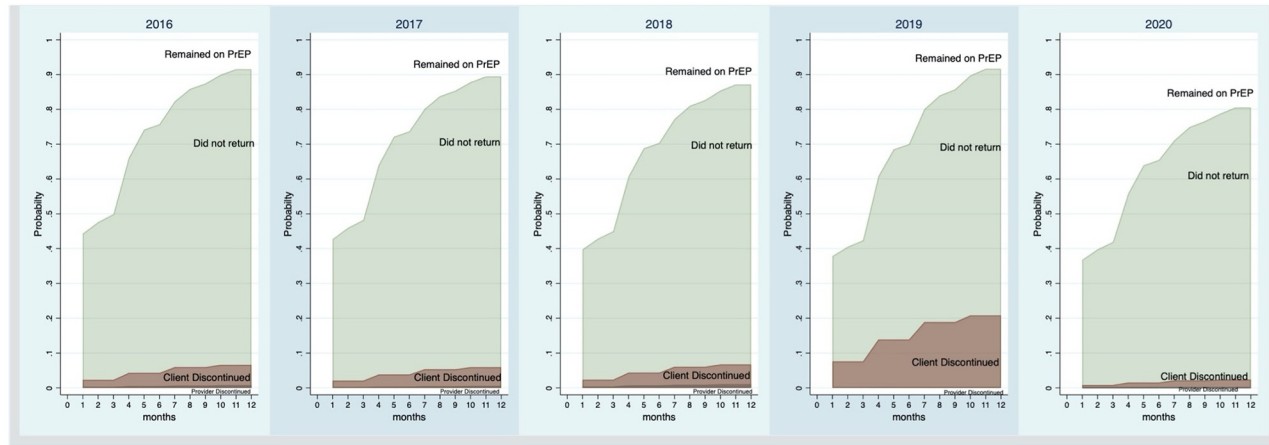

**Fig 2. Stacked cumulative incidence functions plotting the competing risks of those who remained on PrEP, those who did not return (two consecutive missed visits), client-initiated discontinuation, and provider-initiated discontinuation among 2776 female sex workers initiating PrEP through TB HIV Care in eThekwini, South Africa 2016–2020.**

demonstration study with FSW in South Africa also indicated that PrEP use declines to 50% one month after initiation and continues to decline after that [14]. The real-world data presented here reinforce the need for effective implementation strategies to improve PrEP persistence.

Losses were highest among those under 25 years, which is also the group of FSW at the highest risk of HIV acquisition. Nearly 30% of all new infections in South Africa are among young women and by age 25, 70% of FSW have been infected with HIV [24]. Younger women tend to have lower levels of HIV-related knowledge and lower risk perception [25]. Additionally, younger FSW may face specific challenges associated with their youth: less experience in the industry, less social support and fewer network ties, greater vulnerability to violence, lack of financial autonomy, and underdeveloped self-efficacy [26,27]. Strategies to promote PrEP persistence may need to be tailored to the specific challenges and needs of youth.

Over the life of the program, a number of changes have been implemented, both in terms of the goals of the program and the strategies implemented to try to achieve these goals. At the outset of the program, the primary goal was to try to promote PrEP uptake. As the program evolved, PrEP persistence grew to be a more central concern, with CDC/PEPFAR adding current users of PrEP as an official program target in 2019. The shifting priorities of the program may help explain the greater focus given to persistence and the observed improvements over time. Additionally, the program instituted several implementation strategies that aimed to address different barriers to PrEP persistence. The program added staff in June 2019 that included case managers and social workers. These additional staff meant more dedicated resources and greater flexibility to manage new PrEP users. The expanded use of the case management approach allowed for staff to call those who may have missed appointments. This closer follow-up may also help explain the reason for the increased risk of client-initiated discontinuation in 2019, as staff were actually able to ascertain and record that a woman no longer wanted to be on PrEP.

The data presented here show that the program was largely resilient to the interruptions caused by COVID-19, with a consistent increase in the number of initiations and improved persistence among those who began PrEP in 2020. Though empiric data on the impact of COVID-19 on HIV services in Sub-Saharan Africa are limited, most suggest that patient care was interrupted by pandemic restrictions due to lower supply of drugs and fewer patient visits [28–30]. In a national survey in South Africa that recruited approximately 19,000 individuals via social media, it was found that 13.2% were unable to pick up their medications during the pandemic [29]. The implementation strategies started by the program coupled with the fact that HIV prevention and treatment services remained "essential" in South Africa during the COVID-19 pandemic may have made the program more resilient to the shocks of the pandemic.

There are three key limitations of this study. First, the primary outcome in this study is PrEP persistence rather than adherence, which may be better captured through pill counts, electronic monitoring, or pharmacological samples, and arguably is an important measure of whether a woman was protected from acquiring HIV. PrEP persistence is probably a highly specific but not perfectly sensitive measure of PrEP adherence. PrEP was not available from other sources in the study setting during the study period (not persisting on PrEP is a good marker of not adhering to PrEP). Contrariwise, it is unlikely that women would return to pick up PrEP if they did not intend to take it (PrEP persistence is not a guarantee of PrEP adherence, but it is probably a reasonable proxy). If anything, we have overestimated PrEP adherence and our central conclusion (PrEP adherence is too low) is unchanged. A second limitation is that the frequency of PrEP refill visits changed during the study period. At the start of the program, women were seen monthly to be given their PrEP. When the guidelines

were updated to allow for multi-month dispensing, women were seen at 1-month and thereafter 3-monthly (e.g. 1-month, 4-months, 7-months, etc.) to be given their PrEP. In many instances, however, the program did see the client in the intervening months between PrEP refill visits and would document whether an individual was still taking PrEP. While in later years this might have limited our ability to detect PrEP discontinuation with the same speed (e.g., discontinuation that occurred at month 2 might not be detected until month 4,) we chose to retain the definition of two missed monthly visits for discontinuation both to ensure comparability across the years and because the program's register still documents monthly contact with PrEP users. All discontinuations would still be detected within the outcome window (1 year) overall estimates of PrEP persistence at 4- and 7-months should not be affected. As a final limitation, because of limited data availability, demographic and/or behavioral risk data beyond age and year of PrEP initiation were not available to thoroughly assess predictors of persistence. Understanding who is at risk of discontinuation and reasons for discontinuation will be an important contribution to better tailoring strategies to promote PrEP persistence.

The need for PrEP, unlike treatment for HIV, depends heavily on individual risk-assessment and readiness. Some of the improved persistence over time could be explained by program staff better refining who they started on PrEP, that is, better identifying women who were in need of and who understood the benefit of taking PrEP or may in part be due to the role of diffusion of information (increasing PrEP awareness and awareness of the program) and greater acceptability among peers, partners, and others in the community. While this study describes how the program was able to make gradual improvements in PrEP persistence over time while reaching a larger number and proportion of FSW over time, further research on which implementation strategies may have spurred some of these incremental changes and the mechanism of action are critical to see the additional gains in persistence that are needed.

## Supporting information

**S1 Appendix. PrEP eThekwini survival analysis appendix.**
(DOCX)

## Author Contributions

**Conceptualization:** Amrita Rao, Hlengiwe Mhlophe, Carly Comins, Katherine Young, Catherine Lesko, Stefan Baral, Harry Hausler, Sheree Schwartz.

**Data curation:** Hlengiwe Mhlophe, Carly Comins, Mfezi Mcingana, Ntambue Mulumba.

**Formal analysis:** Amrita Rao, Hlengiwe Mhlophe, Catherine Lesko, Sheree Schwartz.

**Funding acquisition:** Amrita Rao, Katherine Young, Stefan Baral, Harry Hausler, Sheree Schwartz.

**Investigation:** Mfezi Mcingana, Sheree Schwartz.

**Methodology:** Amrita Rao, Hlengiwe Mhlophe, Katherine Young, Mfezi Mcingana, Catherine Lesko, Stefan Baral, Harry Hausler.

**Project administration:** Carly Comins, Katherine Young, Mfezi Mcingana, Ntambue Mulumba, Stefan Baral, Harry Hausler.

**Resources:** Hlengiwe Mhlophe, Katherine Young, Ntambue Mulumba, Stefan Baral, Harry Hausler.

**Supervision:** Harry Hausler.

**Validation:** Catherine Lesko, Sheree Schwartz.

**Visualization:** Catherine Lesko.

**Writing – original draft:** Amrita Rao, Hlengiwe Mhlophe, Catherine Lesko, Sheree Schwartz.

**Writing – review & editing:** Hlengiwe Mhlophe, Carly Comins, Katherine Young, Mfezi Mcingana, Catherine Lesko, Ntambue Mulumba, Stefan Baral, Harry Hausler, Sheree Schwartz.

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
