## [Decision Letter · Decision Letter 0]

12 Oct 2021

PONE-D-21-28140Persistence on oral pre-exposure prophylaxis (PrEP) among female sex workers in eThekwini, South Africa, 2016-2020PLOS ONE

Dear Dr. Rao,

Thank you for submitting your manuscript to PLOS ONE. After careful consideration, we feel that it has merit but does not fully meet PLOS ONE’s publication criteria as it currently stands. Therefore, we invite you to submit a revised version of the manuscript that addresses the points raised during the review process.

We look forward to receiving your revised manuscript.

Kind regards,

Deborah Donnell, Ph. D.

Academic Editor

PLOS ONE

Journal Requirements:

2. Please include your tables as part of your main manuscript and remove the individual files. Please note that supplementary tables should remain as separate "supporting information" files.

Additional Editor Comments:

Both reviewers comment on the need for additional details in the methods and results that are necessary to ensure readers have enough context to place the findings of the manuscript alongside findings of others. Please provide the requested details.

Reviewer 1 points out the different definitions of persistence used in the literature and reporting of PrEP uptae. Please ensure the resubmission addresses this, either through revised definitions or through discussion of comparison with other results.

I note that the data will not be available becuase of legal constraints based on the source of the data. I have requested that a PLOS Editor make a determination of that exception.

Reviewers' comments:

Reviewer's Responses to Questions

**Comments to the Author**

1. Is the manuscript technically sound, and do the data support the conclusions?

Reviewer #1: Partly

Reviewer #2: Yes

2. Has the statistical analysis been performed appropriately and rigorously? 

Reviewer #1: No

Reviewer #2: Yes

3. Have the authors made all data underlying the findings in their manuscript fully available?

Reviewer #1: No

Reviewer #2: Yes

4. Is the manuscript presented in an intelligible fashion and written in standard English?

Reviewer #1: Yes

Reviewer #2: Yes

5. Review Comments to the Author

Reviewer #1: INTRO:

1) PrEP persistence instead of persistence on PrEP sounds better and less clumsy

2) Police arrest seems redundant

3) Reference 20 seems to be the wrong reference for line 115. Also the roll out happened with sex workers first - https://www.prepwatch.org/early-lessons-south-africa-rollout/

METHODS:

4) When you looked at PrEP initiation were these all new initiations or did some people stop and then initiate again? Please clarify in the methods how your cohort is defined. Results indicate that some were restarts.

5) There are different ways to define PrEP persistence – you could reference how other people have defined this and why you chose your definition. Please also explain why people had to miss 2 consecutive visits to be defined as “discontinued” It seems to be a more generous definition of persistence and may explain why your persistence is higher than other studies

6) Did you bring your participants back for monthly visits in your program? This is not comparable to most standard of care that would see PrEP clients after a month and then 3 monthly. Please describe the program in the methods and justify why this is “real world implementation” (line 118). Please also discuss how many FSW accessed the program in total year on year and how many of the non infected FSW refused to take PrEP. It is important to have context to be able to understand what the results mean.

7) Some results are reported as “significant” please define in the methods what you mean by this

RESULTS:

8) General comment on the results is that there are no denominators so it is very difficult to interpret your % results.

9) PrEP initiations increased year on year – please include denominators to understand if the project was scaling up or if you were seeing the same numbers of people and more were taking up PrEP. So something like of the xxx FSW eligible to receive PrEP yyy initiated PrEP per year.

10) Please give the n/N for the PrEP persistence figures – need to see the denominators to be able to interpret your results.

11) PrEP persistence was 53% (95% CI: 51%-55%) 1 month after initiation, 33% (95% CI: 31%-35%) 4-months after initiation, and 18% (95% CI: 7-months after initiation. Is this overall? (line 162)

12) Please clarify how you know your results are significant? What statistical test did you use?

DISCUSSION:

13) No limitations mentioned

FIGURES AND LEGENDS

14) Figure 1 legend is incomplete

Reviewer #2: This is a well written and well done study on PrEP persistence in South Africa. While some of the descriptions of persistence and conclusions about age are not new, the study makes a valuable contribution to the literature with interesting findings related to persistence over time and during the pandemic. I enjoyed reading this paper and think it is important.

• Can you describe more about the study population? It is stated that they were already accessing other care at TB HIV care or took part in a PrEP initiation campaign. Would you expect this population of FSW to be representative of FSW more broadly?

• Given the use of register data, please describe more about how you dealt with missing data and measurement error.

• Were reasons for discontinuation captured? This might be interesting to describe by years. For example, was it related to moving, change in behavior or something else.

• Given the way the categories are defined it seems as though there would be differential loss to follow up by group for other reasons unrelated to PrEP discontinuation ( e.g going to a new provider for PrEP) Can you describe how this may have impacted results?

• The authors mention the role of partners and social influence but mostly focus on changes in clinic implementation to explain increases in persistence over time. What about the role of diffusion of information and social norms?

6. PLOS authors have the option to publish the peer review history of their article (what does this mean?). If published, this will include your full peer review and any attached files.

Reviewer #1: No

Reviewer #2: No

---

## [Author Response · Author response to Decision Letter 0]

9 Dec 2021

We thank the reviewers for taking the time to review our manuscript and for your thoughtful and constructive comments. We have responded to your feedback, point-by-point, below. 

Reviewer #1: INTRO:

1) PrEP persistence instead of persistence on PrEP sounds better and less clumsy

Thank you for this suggestion. We have replaced instances of “persistence on PrEP” with “PrEP persistence.”

2) Police arrest seems redundant

We have revised this simply to state “arrest,” to avoid the redundancy.

3) Reference 20 seems to be the wrong reference for line 115. Also the roll out happened with sex workers first - https://www.prepwatch.org/early-lessons-south-africa-rollout/

Thank you very much for this point of clarification. This was an incorrect statement. We have revised it and referenced PrEP Watch instead. 

METHODS: 

4) When you looked at PrEP initiation were these all new initiations or did some people stop and then initiate again? Please clarify in the methods how your cohort is defined. Results indicate that some were restarts. 

This is an important question; thank you for bringing this to our attention. Our study population is made up of unique individuals who initiated PrEP for the first time. Restarts were documented as events occurring during follow-up of these unique individuals in the program’s PrEP register. We have added language to clarify this in the Methods section. 

5) There are different ways to define PrEP persistence – you could reference how other people have defined this and why you chose your definition. Please also explain why people had to miss 2 consecutive visits to be defined as “discontinued” It seems to be a more generous definition of persistence and may explain why your persistence is higher than other studies 

Thank you for this point of clarification. Two consecutive missed months rather than one missed visit was used to define discontinuation, as this is the definition used by the program to define loss to follow-up. If a client misses one visit, but returns the next month, the program notes this client as not attending a visit but continuing on PrEP. If a client misses two consecutive visits and returns subsequently, the program defines her as a “restart.” We have added clarification of this to the Methods section (lines 159-164). Of additional note, this definition is in alignment with the PEPFAR definition used by the Monitoring and Evaluation team and across the PEPFAR funded program in South Africa.

6) Did you bring your participants back for monthly visits in your program? This is not comparable to most standard of care that would see PrEP clients after a month and then 3 monthly. Please describe the program in the methods and justify why this is “real world implementation” (line 118). Please also discuss how many FSW accessed the program in total year on year and how many of the non infected FSW refused to take PrEP. It is important to have context to be able to understand what the results mean. 

Thank you for this question. We agree that further information is needed here to clarify how frequently clients were seen by the program. When the program first began delivering PrEP, clients would be seen on a monthly basis to be given their PrEP. When the guidelines were updated to allow for multi-month dispensing and for women to been seen at 1 month and then again 3-monthly (e.g. 4, 7, 10 months, etc.), the program also updated their procedures accordingly. In many instances, however, the program did see the client in the intervening months between PrEP refill visits and could document whether she was still taking PrEP. 

This could partially explain why for the later years of the program, you see steeper numbers discontinuing at the formal intervals (e.g. 4 months and 7 months) compared with intervening months (e.g. 2 and 3 months), but it is still possible that women may discontinue at intervening months in the later years. 

While this change in frequency of visits is a limitation, we chose to retain the definition of two missed monthly visits for discontinuation both to ensure comparability across the years and because the program’s register still documents monthly contact with PrEP users. 

We have now added detailed reference to this in the Limitations section of the Discussion (lines 317-326).

With regard to the second portion of your question on how many FSW accessed the program, please see the below response to comment # 9; we have added the number of FSW eligible for PrEP (accessing services and HIV-negative) each year as a denominator for the number taking up PrEP. 

7) Some results are reported as “significant” please define in the methods what you mean by this

Results were determined to be statistically significant if the 95% confidence interval around the hazard ratios did not overlap with the line of no difference (which is 1 in the case of ratio measures). We have added a line describing this in the Methods. 

RESULTS: 

8) General comment on the results is that there are no denominators so it is very difficult to interpret your % results. 

Thank you for this. We have provided a table in the Appendix that details the number at risk at the start of each time period, the number who experienced the outcome of discontinuation, the net lost, and the corresponding survivor function. The percentage retained at each monthly interval are based on a Kaplan-Meier survivor function (conditional probability), which accounts for more complex survival data (e.g. adequately accounting for those individuals who were administratively censored: those who initiated in the last few months of the program and did not have enough follow-up months to determine their outcome). We feel that providing denominators would not adequately convey this, and so instead have provided the table to give further detail of the risk sets for each month.

9) PrEP initiations increased year on year – please include denominators to understand if the project was scaling up or if you were seeing the same numbers of people and more were taking up PrEP. So something like of the xxx FSW eligible to receive PrEP yyy initiated PrEP per year. 

We agree with your feedback and have now added into the results the number who were eligible each year for PrEP and the percentage taking up PrEP (lines 194-198). We have also added a new figure (new Figure 1) to visually depict the increase in percentage of FSW initiating PrEP over time.

10) Please give the n/N for the PrEP persistence figures – need to see the denominators to be able to interpret your results. 

Please see our above response to comment # 8. 

11) PrEP persistence was 53% (95% CI: 51%-55%) 1 month after initiation, 33% (95% CI: 31%-35%) 4-months after initiation, and 18% (95% CI: 7-months after initiation. Is this overall? (line 162)

Thank you for this clarifying question. Yes – these results were for overall across all those initiating PrEP.

12) Please clarify how you know your results are significant? What statistical test did you use?

Thank you for this. As we noted above in response to comment # 7, groups were considered to be significantly different from one another if the 95% confidence intervals surrounding the hazard ratio did not overlap with 1. Where we do note “significance” in the results, we have now revised this to state “statistically significant” for added clarity. 

DISCUSSION:

13) No limitations mentioned 

We have now added a limitations paragraph to the Discussion section with the three main limitations of this study (lines 312-330). 

FIGURES AND LEGENDS 

14) Figure 1 legend is incomplete

Thank you for your careful review here. We have now corrected this, so the legend for Figure 1 is complete. 

Reviewer #2: This is a well written and well done study on PrEP persistence in South Africa. While some of the descriptions of persistence and conclusions about age are not new, the study makes a valuable contribution to the literature with interesting findings related to persistence over time and during the pandemic. I enjoyed reading this paper and think it is important.

We thank you for your careful review and consideration. Please find responses to your points below. 

• Can you describe more about the study population? It is stated that they were already accessing other care at TB HIV care or took part in a PrEP initiation campaign. Would you expect this population of FSW to be representative of FSW more broadly?

This is a really important point. Our study population is made up of those accessing services through or engaged with TB HIV Care in some way (from regularly testing for HIV, to picking up contraceptives, to operating out of a venue where TB HIV Care staff are familiar). This population may not be representative of all FSW (those missed may include those working in isolated environments, who are not otherwise interacting with the larger sex work community) but is certainly representative of those who may be reached with outreach services. 

• Given the use of register data, please describe more about how you dealt with missing data and measurement error.

Thank you for this question. Data were complete for our primary independent variables (age and year of initiation). One component of our composite outcome was defined as not attending two consecutive visits, and individuals who missed two consecutive visits were classified as not returning for PrEP. Because of the structure of our outcome, we also did not have missing outcome data. 

• Were reasons for discontinuation captured? This might be interesting to describe by years. For example, was it related to moving, change in behavior or something else.

We agree with your feedback that it would have been informative to present reasons for discontinuation, but unfortunately, the PrEP register we have access to does not capture this information. This is something we are hoping to explore further in future work (there is some ongoing qualitative work in the same population to look at exactly this). 

Additionally, we have now noted in the Limitations section: “As a final limitation, because of limited data availability, demographic and/or behavioral risk data beyond age and year of PrEP initiation were not available to thoroughly assess predictors of persistence. Understanding who is at risk of discontinuation and reasons for discontinuation will be an important contribution to better tailoring strategies to promote PrEP persistence.” 

• Given the way the categories are defined it seems as though there would be differential loss to follow up by group for other reasons unrelated to PrEP discontinuation ( e.g going to a new provider for PrEP) Can you describe how this may have impacted results?

During the study period, TB HIV Care was the sole provider of PrEP to this population, and so it is unlikely that women would have been able to continue their PrEP elsewhere. In this analysis, we have captured three different reasons for PrEP discontinuation: not returning for visits, client-initiated discontinuation, and provider-initiated discontinuation.

• The authors mention the role of partners and social influence but mostly focus on changes in clinic implementation to explain increases in persistence over time. What about the role of diffusion of information and social norms?

Thank you for this point. We have now made note in the Discussion section that changes in persistence over time may in part be due to the role of diffusion of information (PrEP awareness and awareness of the program) and greater acceptability among peers, partners, and others in the community.

---

## [Decision Letter · Decision Letter 1]

2 Mar 2022

Persistence on oral pre-exposure prophylaxis (PrEP) among female sex workers in eThekwini, South Africa, 2016-2020

PONE-D-21-28140R1

Dear Dr. Rao,

We’re pleased to inform you that your manuscript has been judged scientifically suitable for publication and will be formally accepted for publication once it meets all outstanding technical requirements.

Kind regards,

Catherine E Oldenburg

Academic Editor

PLOS ONE

Additional Editor Comments (optional):

Reviewers' comments:

Reviewer's Responses to Questions

**Comments to the Author**

1. If the authors have adequately addressed your comments raised in a previous round of review and you feel that this manuscript is now acceptable for publication, you may indicate that here to bypass the “Comments to the Author” section, enter your conflict of interest statement in the “Confidential to Editor” section, and submit your "Accept" recommendation.

Reviewer #1: All comments have been addressed

Reviewer #2: All comments have been addressed

2. Is the manuscript technically sound, and do the data support the conclusions?

Reviewer #1: (No Response)

Reviewer #2: Yes

3. Has the statistical analysis been performed appropriately and rigorously? 

Reviewer #1: (No Response)

Reviewer #2: Yes

4. Have the authors made all data underlying the findings in their manuscript fully available?

Reviewer #1: (No Response)

Reviewer #2: No

5. Is the manuscript presented in an intelligible fashion and written in standard English?

Reviewer #1: (No Response)

Reviewer #2: Yes

6. Review Comments to the Author

Reviewer #1: (No Response)

Reviewer #2: The authors have addressed my comments. I have no additional comments for the authors and agree that it should be accepted.

7. PLOS authors have the option to publish the peer review history of their article (what does this mean?). If published, this will include your full peer review and any attached files.

Reviewer #1: No

Reviewer #2: No

---

## [Editor Report · Acceptance letter]

7 Mar 2022

PONE-D-21-28140R1 

Persistence on oral pre-exposure prophylaxis (PrEP) among female sex workers in eThekwini, South Africa, 2016-2020 

Dear Dr. Rao:

I'm pleased to inform you that your manuscript has been deemed suitable for publication in PLOS ONE. Congratulations! Your manuscript is now with our production department. 

Kind regards, 

on behalf of

Dr. Catherine E Oldenburg 

Academic Editor

PLOS ONE